# DNA methylation presents distinct binding sites for human transcription factors

**Shaohui Hu[1,2†‡], Jun Wan[3†], Yijing Su[4,5†], Qifeng Song[1,2], Yaxue Zeng[4,5], Ha Nam Nguyen[4,6], Jaehoon Shin[4,6], Eric Cox[1,2], Hee Sool Rho[1,2], Crystal Woodard[1,2], Shuli Xia[5,7], Shuang Liu[8], Huibin Lyu[8], Guo-Li Ming[4,5,6,9], Herschel Wade[10]\*, Hongjun Song[4,5,6,9]\*, Jiang Qian[3]\*, Heng Zhu[1,2]\***

[1]Department of Pharmacology and Molecular Sciences, Johns Hopkins University School of Medicine, Baltimore, United States; [2]Center for High-Throughput Biology, Johns Hopkins University School of Medicine, Baltimore, United States; [3]The Wilmer Eye Institute, Johns Hopkins University School of Medicine, Baltimore, United States; [4]Institute for Cell Engineering, Johns Hopkins University School of Medicine, Baltimore, United States; [5]Department of Neurology, Johns Hopkins University School of Medicine, Baltimore, United States; [6]Cellular and Molecular Medicine Graduate Program, Johns Hopkins University School of Medicine, Baltimore, United States; [7]Hugo W Moser Research Institute at Kennedy Krieger, Johns Hopkins University School of Medicine, Baltimore, United States; [8]Institute of Physics, Chinese Academy of Sciences, Beijing, China; [9]The Solomon Snyder Department of Neuroscience, Johns Hopkins University School of Medicine, Baltimore, United States; [10]Department of Biophysics and Biophysical Chemistry, Johns Hopkins University School of Medicine, Baltimore, United States

**\*For correspondence:** herschel.
wade@jhmi.edu (HW); shongju1@
jhmi.edu (HS); jiang.qian@jhmi.edu
(JQ); hzhu4@jhmi.edu (HZ)

†These authors contributed
equally to this work

‡Present address: CDI
Laboratories, Baltimore, United
States

**Competing interests:** The
authors declare that no
competing interests exist.

**Reviewing editor**: Danny
Reinberg, New York University
School of Medicine, United
States

**Abstract** DNA methylation, especially CpG methylation at promoter regions, has been generally considered as a potent epigenetic modification that prohibits transcription factor (TF) recruitment, resulting in transcription suppression. Here, we used a protein microarray-based approach to systematically survey the entire human TF family and found numerous purified TFs with methylated CpG (mCpG)-dependent DNA-binding activities. Interestingly, some TFs exhibit specific binding activity to methylated and unmethylated DNA motifs of distinct sequences. To elucidate the underlying mechanism, we focused on Kruppel-like factor 4 (KLF4), and decoupled its mCpG- and CpG-binding activities via site-directed mutagenesis. Furthermore, KLF4 binds specific methylated or unmethylated motifs in human embryonic stem cells in vivo. Our study suggests that mCpG-dependent TF binding activity is a widespread phenomenon and provides a new framework to understand the role and mechanism of TFs in epigenetic regulation of gene transcription.

## Introduction

DNA methylation is an ancient and major epigenetic modification that plays an important role in key biological processes, including genomic imprinting, X-chromosome inactivation, suppression of transposable elements, and carcinogenesis (*Jaenisch and Bird, 2003*; *Egger et al., 2004*; *Robertson, 2005*; *Feinberg, 2007*; *Reik, 2007*). In higher eukaryotes, methylation of CpG sites, especially at promoter regions, is generally considered as the hallmark of gene silencing (*Baylin, 2005*). The molecular consequence of CpG methylation is generally believed to disrupt TF–DNA interactions either directly (*Nan et al., 1998*), or indirectly by recruiting sequence-independent methylated DNA-binding proteins that occupy the methylated promoters and compete for the TF binding sites (*Boyes and Bird, 1991*). So far,

**eLife digest** DNA methylation—the addition of a methyl group to a cytosine or adenine base within DNA—has a key role in regulating the expression of genes as proteins. It contributes to processes such as X-inactivation, in which one copy of the X chromosome is silenced in females, and genomic imprinting, in which the expression of a gene depends upon which parent it was inherited from. DNA methylation has also been implicated in the development of cancer. However, the molecular mechanisms by which it produces these effects are not fully understood.

In mammals, the methylation of CpG sites—which consist of a cytosine base next to a guanine base—is typically thought to reduce gene expression by preventing proteins called transcription factors from binding to regions of DNA called promoters. This can occur directly if methylation disrupts interactions between the DNA and the transcription factors, or indirectly if other proteins that bind to the methylated DNA compete with the transcription factors for binding sites. However, only a small number of proteins that bind to methylated DNA have so far been identified.

Now, Hu et al. have screened the entire family of roughly 1300 human transcription factors and 210 co-factors (proteins that interact with transcription factors) for their ability to bind to some 150 different stretches of methylated DNA. They found that 47 of the proteins could bind to methylated CpG sites, with the majority showing a preference for specific DNA sequences. Moreover, some transcription factors and co-factors bind to methylated and non-methylated DNA targets with distinct sequences. These two types of binding are largely independent, as illustrated by the fact that mutations that prevent a transcription factor called KLF4 from binding to methylated DNA do not prevent it binding to unmethylated DNA, and vice versa.

The work of Hu et al. suggests that methylated cytosine can effectively act as a 'fifth base'—in addition to adenine, cytosine, guanine and thymine—and emphasizes the importance of DNA methylation for regulating gene expression.

only MeCP2, MBD1, MBD2, and a few zinc finger proteins have been identified as bona fide methylated DNA-binding proteins (*Lewis et al., 1992*; *Meehan et al., 1989*; *Filion et al., 2006*; *Bartke et al., 2010*; *Bartels et al., 2011*; *Quenneville et al., 2011*; *Spruijt et al., 2013*). It is unclear whether the methylated DNA binding activity is widespread among different TF subfamilies. Furthermore, the transcriptional regulatory activity of these methylation-dependent TF–DNA interactions has not been explored. Finally, the structural basis of these methylation-dependent TF–DNA interactions remains elusive.

Here we employed a protein microarray-based approach to characterize the entire human TF repertoire for their direct binding capacity to a large set of methylated DNA motifs. In this work, we simultaneously tested the methylated DNA-binding activity for the majority of human TFs. Since we examined each methylated DNA motif individually on the protein microarray, we were able to determine the specific sequences surrounding the methylated CpG that is recognized by TFs. Furthermore, we demonstrated that the methylated- and unmethylated-DNA binding activities of KLF4 could be decoupled by protein mutagenesis.

## Results

We selected a total of 154 DNA motifs (*Supplementary file 1A*) with the following characteristics: (1) predicted to be potential TF-binding sites in promoter regions of the human genome (*Xie et al., 2005*; *Yu et al., 2006*; *Elemento et al., 2007*); (2) representative of a subset of the 460 DNA motifs that have been examined for protein-binding activity in our previous study (*Hu et al., 2009*; *Xie et al., 2010*); and (3) carrying at least one CpG site. Because our goal was to identify DNA methylation-dependent binding activity, we developed a competition assay on a protein microarray to identify TFs that prefer DNA motifs carrying mCpGs (*Figure 1A*). Each synthesized double-stranded DNA motif was end-labeled with Cy5 (*Hu et al., 2009*) and converted to the double-stranded mCpG form using a bacterial CpG DNA methylase *Sss*I. Each methylated motif was incubated with the protein microarray in the presence of its unlabeled (cold), unmethylated counterpart in 10-fold excess. The competition assay was first tested on a pilot array, comprised of MBD1, MBD2, MeCP2, and several negative control proteins. As illustrated in *Figure 1B*, strong binding signals were observed in a protein concentration-dependent manner,

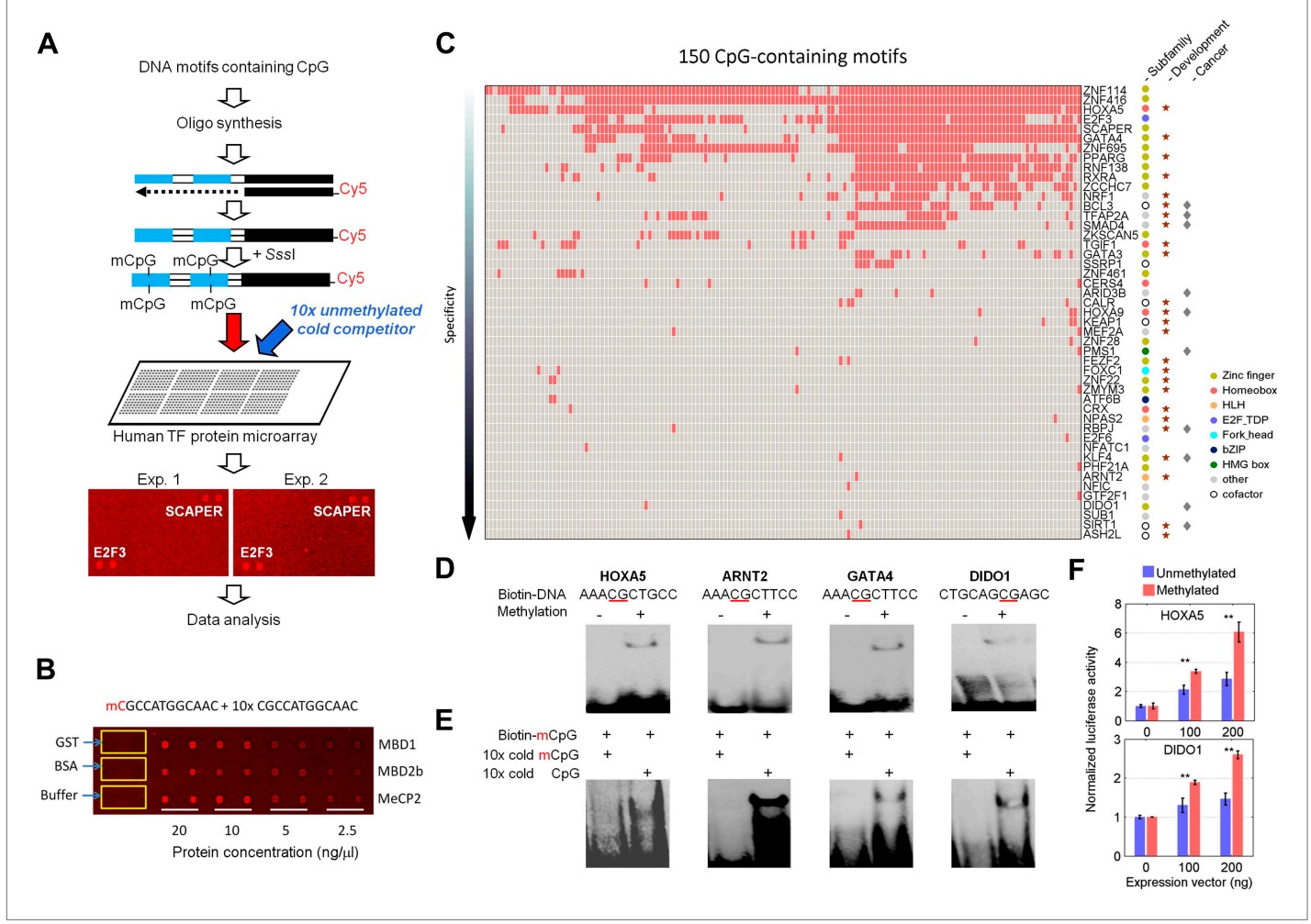

**Figure 1**. Protein microarray-based approach identified mCpG-dependent DNA-binding activity among human TFs and cofactors. (**A**) A competition assay was used to identify proteins that preferentially bind to methylated DNA motifs. SCAPER (S-phase cyclin A-associated protein in the ER) and E2F3 (E2F transcription factor 3) were shown here as two examples of methylated DNA-binding proteins. (**B**) A proof-of-principle assay was conducted using known methylated DNA-binding proteins on a pilot protein microarray. (**C**) Binding profiles of 41 TFs and 6 co-factors against 150 of the 154 tested methylated DNA motifs are summarized in the interaction map. TFs are color-coded based on the subfamilies. (**D**) EMSA assays validated DNA-binding activity for four selected TF candidates. Representative images from three independent experiments with similar results are shown. (**E**) Competition EMSA assays confirmed mCpG-dependent DNA-binding activities. As expected, 10-fold unlabeled, methylated DNA motif readily abolished the protein–DNA complex formation of the tested TFs with the biotinylated and methylated DNA motifs (Lane 1 in each image). However, 10-fold cold unmethylated DNA counterparts could not compete off methylated DNA binding, consistent with the protein microarray results. (**F**) HOXA5 and DIDO1 showed mCpG-dependent activation of luciferase activity in GT1-7 cells. Values represent mean ± SD (n = 3; **: p<0.01; *t*-test).

The following figure supplements are available for figure 1:

**Figure supplement 1**. Data analysis of the protein microarray assays.

**Figure supplement 2**. Reproducibility of protein microarray data.

**Figure supplement 3**. Distribution of number of mCpG-binding TFs/co-factors in a given motif-bind assay.

**Figure supplement 4**. Distribution of number of methylated motifs recognized by a given TF/co-factor.

**Figure supplement 5**. Distribution of TF subfamily members.

*Figure 1. Continued on next page*

*Figure 1. Continued*

**Figure supplement 6**. Four additional EMSA assays (A) and competition EMSA assays (B).

**Figure supplement 7**. Methylation level of the KLF4 and HOXA5 luciferase reporter constructs.

**Figure supplement 8**. Number of unique mCpG-binding TFs/co-factors in function of number of tested methylated DNA motifs.

while control proteins, such as GST and BSA, did not show any detectable signals, confirming the reliability of the competition assay.

We then probed for direct interactions between 154 methylated motifs and the human TF repertoire using the TF protein microarray, which includes 1321 TF and 210 co-factor proteins (*Supplementary file 1B*). These full-length human proteins were overexpressed and individually purified in yeast as N-terminal GST fusion proteins as previously reported (*Hu et al., 2009*). The quantity and purity of the purified proteins were examined with Coomassie and silver stains against a random set of purified proteins (data not shown). Each of the labeled 154 methylated motifs was separately incubated on the human TF protein microarray in the presence of the unlabeled counterparts in 10-fold excess. Each reaction was performed in duplicate to ensure reproducibility. After a washing step, binding signals were acquired and processed using the GenePix software (*Figure 1—figure supplement 1*). After the binding signals were further normalized by correcting local background signals with a 9 × 9 sub-grid, Z-scores of each protein on the array were determined as previously described (*Hu et al., 2009*). A Z-score of 3 was used as the cutoff to identify the positives. Correlation analysis of the binding signals showed high reproducibility between each duplicated assay (*Figure 1—figure supplement 2*). Of the 154 methylated motifs examined, 150 (97%) showed significant binding signals to at least one protein on the microarrays (median = 8), suggesting binding to methylated DNA is prevalent, at least in this in vitro competition assay (see 'Materials and methods', *Figure 1—figure supplement 3*). In addition, 41 TFs and 6 TF co-factors (3% of all factors examined) showed mCpG-dependent binding activity (*Figure 1C*). Among these, 15% showed a broad binding activity to over 50% of all methylated motifs tested, suggesting that these proteins are not sensitive to the sequence context surrounding the mCpG site. For example, two zinc finger TFs, ZNF114 and ZNF416, could recognize almost all of the motifs tested in the competition assay. However, the majority of the 47 mCpG-binding proteins showed mCpG- and sequence-dependent binding activity, with 22 TFs exhibiting binding signaling to fewer than three motifs (*Figure 1—figure supplement 4*). Notably, the mCpG binding activity is widespread among various TF subfamilies, such as the zf-C2H2, Homeobox, bHLH, Forkhead, bZIP, and HMG box subfamilies (*Figure 1C*); no TF subfamily was found significantly enriched in the hit list (*Figure 1—figure supplement 5*). On the other hand, consistent with the important role of DNA methylation in the development of cancer (*Egger et al., 2004*; *Reik, 2007*), there was a significant enrichment for known oncogenes and tumor suppressors (10/47; p<0.01; hypergeometric model) and factors involved in tissue development (25/47; p<0.015; hypergeometric model; *Figure 1C*). Together, these results suggest that methylation-dependent direct TF–DNA interaction is a widespread phenomenon among various TF subfamilies in humans.

To validate the protein microarray results, we selected 11 TFs with different mCpG-dependent DNA-binding behaviors (i.e., sequence-dependent or -independent). Among them, six (e.g., ARNT2) bound to fewer than three methylated motifs, while the others (e.g., HOXA5) bound to at least 66 motifs (*Supplementary file 1C*). Using an electrophoretic mobility shift assay (EMSA), we confirmed that 8 of the 11 tested TFs could readily form a protein–DNA complex with their corresponding methylated motifs, but not with the unmethylated form (*Figure 1D*; *Figure 1—figure supplement 6*). This result indicates either a 27% of false positive rate of the protein microarray assay or a higher sensitivity of protein microarray than EMSA. While the binding activities to the mCpG-carrying motifs could be readily competed off with the unlabeled mCpG-containing analogs in 10-fold excess, the unmethylated counterparts showed no obvious impact, indicating that the complex formation between these TFs and specific DNA motifs is not non-specific but requires CpG methylation (*Figure 1E*).

To determine whether mCpG-dependent TF–DNA interactions could convey transcriptional activity in vivo, we selected HOXA5 and DIDO1 to perform cell-based luciferase assays. Motifs M305 and M24 recognized by HOXA5 and DIDO1, respectively, were separately cloned into the promoter region of a

CpG-free luciferase construct (pCpGL) (*Klug and Rehli, 2006*), which was then methylated with *Sss*I and transfected with the *HOXA5* and *DIDO1* expression constructs at different concentrations into a mammalian cell line. The complete methylation of the motifs before transfection was confirmed using Sanger-bisulfite sequencing (*Figure 1—figure supplement 7*). Unmethylated luciferase construct was used as a negative control. Comparing with the luciferase activity of the negative controls, we found that both HOXA5 and DIDO1 demonstrated dose- and methylation-dependent enhancement of the luciferase activity (*Figure 1F*), indicating that the two TFs could specifically bind to the methylated promoters and enhance the downstream gene transcription.

We then asked whether these mCpG-binding TFs also bound to the same consensus motifs in an unmethylated form. Since our previous genome-wide analysis of human TF–DNA interactions included these 150 unmodified CpG-containing motifs (*Hu et al., 2009*), we integrated both protein microarray datasets. We discovered that 17 TFs recognize both methylated and unmethylated motifs (*Figure 2A*). Of the 435 protein–DNA interactions (PDIs) examined, only a small fraction (4%) exhibited motif-binding activity irrespective of the CpG methylation status (*Figure 2A*; yellow bars). Even in these cases, TF binding to methylated forms is much stronger than to the unmethylated forms based on the inability of non-methylated analogs to compete effectively even when present in 10-fold excess. Interestingly, 321 (74%) and 97 (22%) PDIs exhibit specificity to methylated (*Figure 2A*; red bars) and unmethylated motifs (*Figure 2A*; blue bars), respectively. To evaluate sequence specificity for the newly discovered methylation-dependent PDIs, we determined the consensus methylated motif sequences for the 17 TFs. The comparison with their known consensus sequences indicated that, in most cases, the sequences of methylated and unmethylated motifs recognized by the same TFs are distinct (*Figure 2A*; right panel; 'Materials and methods'). For example, nuclear factor I/C (NFIC) binds to a methylated motif M209, GTmCpGCC, while its known consensus sequence is TTGGC, suggesting that many TFs are of dual-specificity, recognizing methylated and unmethylated DNA motifs of distinct sequences.

To experimentally confirm this apparent dual-specificity, we performed EMSA assays with two motifs, recognized by the same TF, in both methylated and unmethylated forms (*Figure 2B*). For instance, KLF4 could form a protein–DNA complex with the methylated motif M197 (TCCmCpGCCC), but not with its unmethylated form (*Figure 2B*). In contrast, methylation on motif M412 (GCTTTTACG) disrupted its interaction with KLF4 (*Figure 2B*). The same phenomenon was confirmed for three additional TFs, namely TFAP2A, ARID3B, and ZMYM3 (*Figure 2B*).

These results raised an interesting question: does binding by a dual-specificity TF to one motif affect binding to the other of different methylation status? By performing a competition EMSA assay we observed two scenarios for different TFs (*Figure 2C–F*). In the first scenario, binding to one motif interferes the binding activity to the other motif. For example, the formation of protein–DNA complex between ARID3B and a methylated motif M319 (AAAmCpGCTTCC) could be readily competed off with the unmethylated motif M47 (GTGGGCGAAA) in 10-fold excess, and vice versa (*Figure 2E*), suggesting that ARID3B either uses the same DNA-binding domain to interact with both motifs, or binding to one motif inhibits binding to the other, presumably via conformational changes or steric hindrance (*Figure 2C*). The specificity of these assays was further confirmed with a competition EMSA assay with ARID3B and ZMYM3 (*Figure 2—figure supplement 1*). In the second scenario, binding activities of a dual-specificity TF are independent of each other (*Figure 2D*). For instance, KLF4–DNA complex formation with the methylated motif M197 was not affected by adding unmethylated motif M412 in 10-fold excess in the competition assay, suggesting that it may use two different domains to distinguish methylated from unmethylated motifs (*Figure 2F*; left panel). As expected, addition of 10-fold excess of methylated M197 could readily compete off the complex formation of KLF4 with labeled M197 (*Figure 2F*; left panel). The same results were observed for TFAP2A (*Figure 2F*; right panel). The binding specificities of KLF4 and TFAP2A were further confirmed by adding both methylated and non-methylated competitor DNA, resulting in disruption of the complex formation (*Figure 2—figure supplement 2*). To facilitate direct comparison, we reproduced all combinations of the competition EMSA assays for KLF4 in parallel on the same gel. The results confirmed its dual-specificity in a non-competitive fashion (*Figure 2—figure supplement 3*). The fact that KLF4 could form protein–DNA complexes with both motifs equally well, regardless of the presence of the competitors, also suggested that it is highly unlikely that these two motif sequences could bind to the same domain at different affinities.

We next employed an oblique incidence reflectivity difference (OIRD) system (*Landry et al., 2004*; *Zhu et al., 2007*; *Fei et al., 2011*) to determine binding affinity (i.e., $K_D$ values) of ZMYM3, TFAP2A,

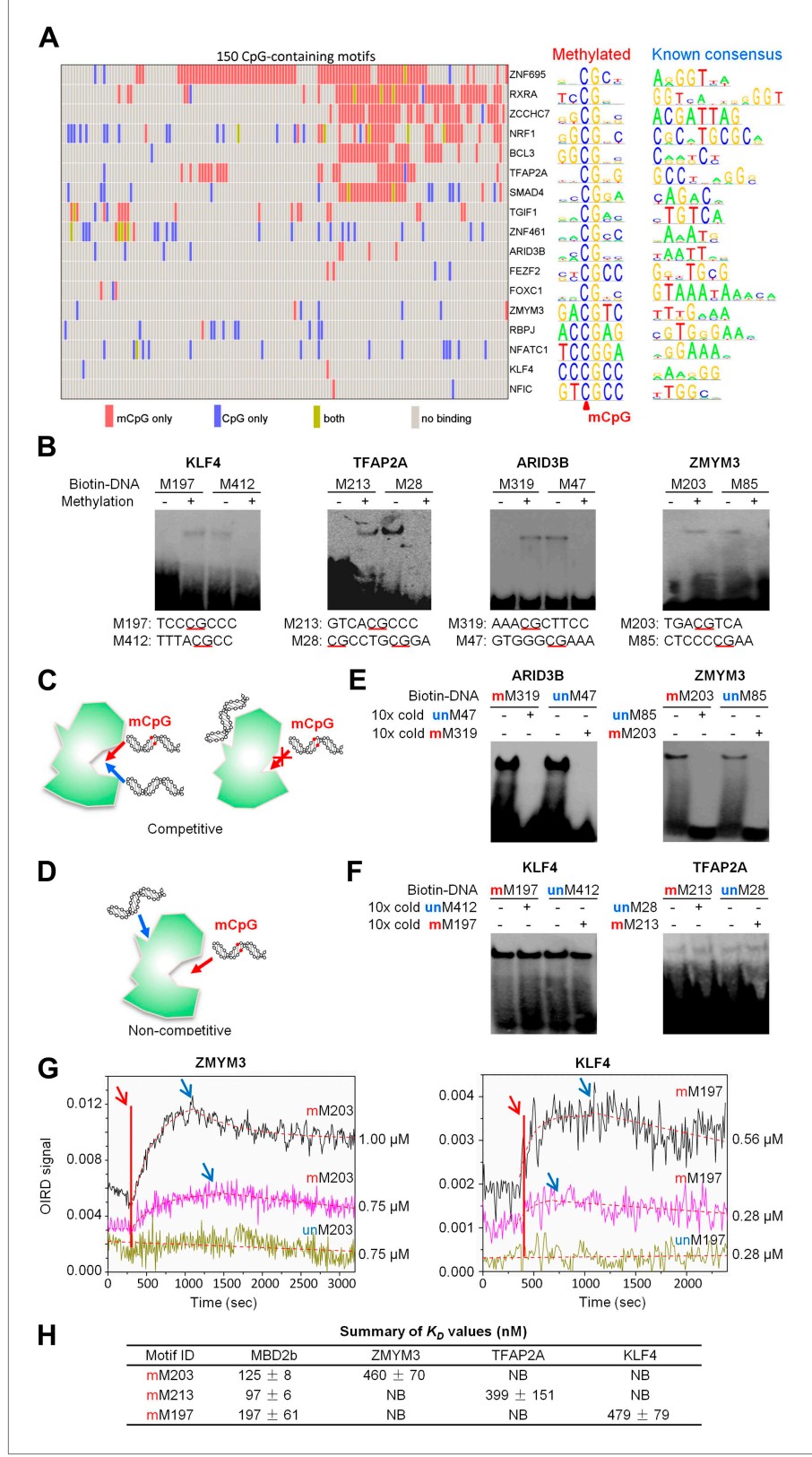

**Figure 2**. A group of 17 TFs can bind to both methylated and unmethylated motifs of distinct sequences. (**A**) Our previous PDI dataset was compiled with the dataset in this study to generate binding preference of the 17 TFs. Methylated consensus motifs of the 17 TFs identified based on the protein microarray results are compared with
*Figure 2. Continued on next page*

*Figure 2. Continued*

their known consensus motifs. (**B**) EMSA assays confirmed that four TFs could specifically interact with both methylated and unmethylated motifs of distinct sequences. Representative images from three independent experiments with similar results are shown. (**C**) and (**D**) Two possible scenarios are proposed to distinguish the mode of interactions between these TFs and their corresponding motifs. (**E**) and (**F**) Competition EMSA assays showed that both scenarios are possible. Representative images from two independent experiments with similar results are shown in each panel. (**G**) OIRD sensorgrams for ZMYM3 and KLF4 binding to methylated motifs M203 and M197, and their unmethylated counterparts, respectively. The OIRD measurements were performed at two concentrations of each protein. Solid lines represent the OIRD signals. Dashed lines are fitted On- and Off-curves. Red arrows indicate the starting point when a TF protein was introduced to the OIRD reaction chamber. Blue arrows indicate the time points when wash buffer was added. (**H**) Summary of average $K_D$ values measured at two concentrations of each protein. 'NB' indicates no observed binding signals.

The following figure supplements are available for figure 2:

**Figure supplement 1**. Competition EMSA assays for ARID3B and ZMYM3.

**Figure supplement 2**. Competition EMSA assays for KLF4 and TFAP2A.

**Figure supplement 3**. Summary of KLF4's dual-specificity.

**Figure supplement 4**. OIRD sensorgrams for three TFs and MBD2b binding to three methylated DNA motifs.

and KLF4 with their corresponding methylated DNA motifs M203, 213, and M197, respectively. As a comparison, the well known methylated DNA-binding protein, MBD2b, was also included. Because the OIRD system can monitor binding events in a real-time, label-free fashion, we therefore obtained the $K_{on}$ and $K_{off}$ values, and determined the $K_D$ values of ZMYM3, TFAP2A, and KLF4 as 460 nM, 399 nM, and 479 nM, respectively (*Figure 2G*; *Figure 2—figure supplement 4A–D*). As expected, none of them showed any significant binding activity to the unmethylated DNA motifs in the OIRD measurements, confirming our previous observations. On the other hand, the $K_D$ values of MBD2b measured against the same motifs are in close range to the three TFs tested (*Figure 2—figure supplement 4A*), suggesting that these TFs could bind to methylated DNA motifs almost as well as MBD2b (*Figure 2H*).

To further dissect the molecular mechanism of this phenomenon, we focused on KLF4 to identify the key residue(s) responsible for interacting with the methylated cytosine, in order to decouple its dual-specificity. Intriguingly, KLF4 encodes a classic zf-C2H2 domain, a truncated and a full zf-H2C2 domain at its very C-terminus (*Figure 3A* and *Figure 3—figure supplement 1*). Because the X-ray structure of KLF4 bound to methylated CpG sequences is not available, we used a modeling approach. These efforts were facilitated by crystal structures of other, unrelated 5mC-binding proteins (e.g., MeCP2 and ZFP57) (*Ho et al., 2008*; *Liu et al., 2012*), as well as the adduct composed of mouse KLF4 and unmethylated DNA, which was employed as a modeling template. Using the existing KLF4 structure (*Schuetz et al., 2011*), we identified Arg458 and Asp460 in the last H2C2 zinc finger as a candidate 5mCpG recognition motif. In the KLF4 model, DNA binding is stabilized by a 5mC-Arg-G via improved van der Waals interaction and an Arg458-Asp460 salt-bridge (*Figure 3A*). Binding is further stabilized by a CH•••O ($H_2$O-5mC) H-bond contact of the methylated cytosine on the complementary strand (*Figure 3B*). Interestingly, the local atomic environment of the KLF4 model is rather similar to those of MeCP2 and ZFP57 in their mCpG-bound forms (*Figure 3—figure supplement 2*). Therefore, Arg458 and Asp460 were predicted to be crucial for KLF4 and mCpG-dependent interaction (*Figure 3—figure supplement 1*).

To test our prediction, we generated a series of KLF4 mutants, including three single (i.e., R458A, R458K, D460A), one double (R458A::D460A) mutants, and one truncation (Δ432). EMSA assay with purified KLF4 mutant proteins showed that all point mutations at either or both residues completely abolished KLF4's binding activity to methylated motif M197 (CCmCpGCC), while none of them had any detectable impact on the binding activity to the unmethylated motif M412 (TACpGCC; *Figure 3C* and *Figure 3—figure supplement 3*). The R458A and R458K mutations showed similar impact on mCpG-dependent binding activity, suggesting that charge on the residue is not critical. Similarly, KLF4 truncation showed no binding activity to methylated motif M197 (CCmCpGCC), but still maintained its

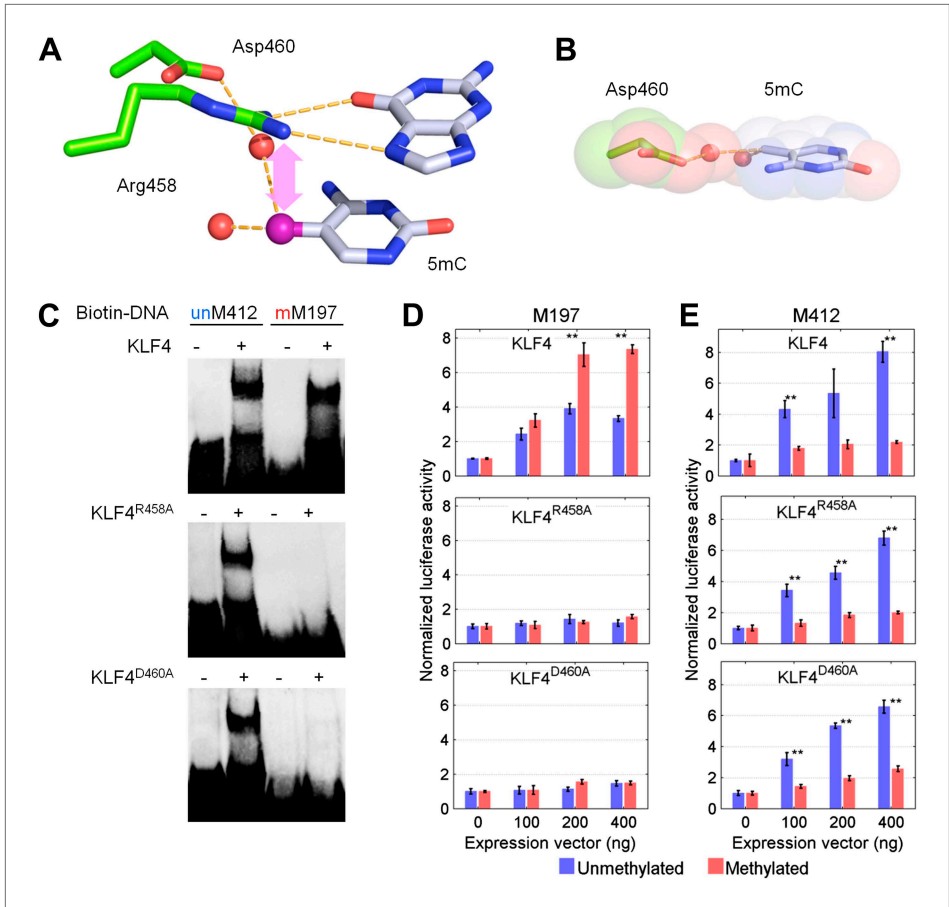

**Figure 3**. KLF4's mCpG-dependent binding activity is decoupled from its binding activity to unmethylated motifs. (**A**) Simulation of KLF4–DNA interactions predicted that two residues, Arg458 and Asp460, are involved in the interactions with methylated cytosine. Double arrow indicates van der Waals interactions between Arg458 and methyl group on the cytosine in one strand (5mC$^A$). Red balls represent water molecules. (**B**) Asp460 further stabilizes binding to 5 mC on the other strand (5mC$^B$) via a CH•••O ($H_2$O—5mC) H-bond contact. (**C**) EMSA assays using KLF4 mutated proteins demonstrated that both R458 and D460 are crucial for mCpG-dependent binding activity. Representative images from three independent experiments with similar results are shown. (**D**) In cell-based luciferase assays for M197, WT KLF4 showed mCpG-dependent activation of downstream gene expression (red bars in the upper panel), while both R458A and D460A mutations abolished this activity (red bars in the middle and lower panels). (**E**) In cell-based luciferase assays with M412 (blue bars), both WT and mutants can activate the expression of unmethylated M412 (blue bars), but have no effect on methylated M412 (red bars). In (**D**) and (**E**), values represent mean ± SD (n = 3; **: p<0.01; *t*-test)

The following figure supplements are available for figure 3:

**Figure supplement 1**. Architecture of KLF4 DNA-binding domain.

**Figure supplement 2**. Known crystal structures of MeCP2 and ZFP57 in complex with methylated DNA.

**Figure supplement 3**. EMSA assays to evaluate impacts of KLF4 R458K, R458A::D460A mutations, and Δ432 truncation on its binding activity to motifs M412 and M197.

**Figure supplement 4**. Western blot analysis of overexpression of KLF4$^{WT}$, KLF4R$^{458A}$ and KLF4D$^{460A}$ proteins in GT1-7 cells.

binding activity to the unmethylated motif M412 (TACpGCC; *Figure 3—figure supplement 3*), indicating that the C2H2 zinc finger in KLF4 mediates KLF4's interaction with unmethylated motif M412.

To examine whether KLF4's binding activity to M197 (CCmCpGCC) is important for transcription regulation, we employed cell-based luciferase assays using both wild-type (WT) and mutated KLF4

expression constructs. Indeed, when co-transfected with the pCpGL construct carrying methylated motif M197 in the promoter, WT KLF4 showed dose-dependent increase of the luciferase activity (*Figure 3D*). This enhancement of transcription is also methylation-dependent, because WT KLF4 could not increase luciferase activity when motif M197 was not methylated in the construct (*Figure 3D*). More importantly, both *KLF4^{R458A}* and *KLF4^{D460A}* constructs completely lost their ability to modulate transcription from the pCpGL construct carrying methylated motif M197; whereas they both showed dose-dependent enhancement of gene transcription in the luciferase assay, like WT KLF4, when co-transfected with the pCpGL construct carrying unmethylated motif M412 (*Figure 3E*). Consistent with the EMSA results, neither WT nor mutant KLF4 constructs showed any significant increase of transcription when motif M412 was methylated in the construct (*Figure 3E*). Western blot analysis demonstrated an equal transfection efficiency of WT *KLF4*, *KLF4^{R458A}* and *KLF4^{D460A}* constructs (*Figure 3—figure supplement 4*). Altogether, these results suggested that KLF4 could bind to methylated and unmethylated CpG sites in different sequence contexts, resulting in activation of downstream gene expression in heterologous cells in vivo, and that these dual specificities are independent and achieved via two functionally separable domains encoded by KLF4.

Finally, we asked whether KLF4 binding preferences to both methylated and unmethylated motifs that we identified in vitro reflect sequences preferred in vivo. Since KLF4 is well known to play an important role in embryonic stem cell (ESC) maintenance, we therefore examined whether mCpG-dependent KLF4–DNA interactions occur in human ESCs (H1). First, we performed bioinformatics analysis of in vivo KLF4-binding sites in H1 cells based on their methylation status (*Figure 4—figure supplement 1*). We superimposed the published in vivo KLF4 binding sites, as determined by the ChIP-seq approach (*Lister et al., 2009*), with the published methylome dataset of single-base resolution obtained with whole-genome bisulfite sequencing in H1 hESCs (*Lister et al., 2009*). We identified numerous cases in which KLF4 tends to bind to those genomic regions containing a methylated CCCpGCC sequence, consistent with our protein microarray results (*Figure 4—figure supplement 1*; lower panel). Among the KLF4-binding sites containing at least one CpG, we found that the majority of these sites could be categorized into two groups across the whole-genome: 48% have high methylation levels of over 80% and 38% have methylation levels lower than 20% (*Figure 4A*). We identified statistically significant consensus motifs in methylated group ('Materials and methods'; *Supplementary file 1D*). The top consensus motif, CmCpGC, discovered in the high methylation group is embedded in motif M197 (CCmCpGCC) that was recognized by KLF4 in the protein microarray assays (*Figure 4A*). Taken together, these integration analyses demonstrate that KLF4's preference to methylated consensus sequence identified in vivo is highly similar to that determined in vitro.

To provide direct experimental evidence that KLF4 binds to endogenous CCmCpGCC in H1 hESCs in vivo, we performed KLF4 ChIP coupled with bisulfite sequencing (*Figure 4B*), using primer pairs designed on the basis of the above bioinformatics analysis. First, chromatin IP (ChIP) demonstrates specific and direct binding of KLF4 to five selected regulatory loci based on KLF4 ChIP-seq signals (*Figure 4C*; *Figure 4—figure supplement 2*). Next, quantitative real-time PCR (q-PCR) also showed the high enrichment (≥9-fold) on the specific loci after KLF4 ChIP (*Figure 4D*). KLF4 ChIP-bisulfite sequencing analyses showed that KLF4 could readily ChIP those loci (i.e., L1 and L2 in *Figure 4E*) with TACpGCC at two completely unmethylated fragments, and with CCCpGCC at ~100% methylation level in other two loci tested (L3 in *Figure 4E*, and *Figure 4—figure supplement 3*), indicating that KLF4 binds to both unmethylated and highly methylated CpGs in different sequence contexts in vivo. To rule out the possibility that other sites within the ChIP'ed fragments might be responsible for KLF4 binding, we searched the published dataset (*Lister et al., 2009*) and did not find any other KLF4 binding peaks within the surrounding 500 bp, supporting that KLF4 indeed binds to these unmethylated or methylated sites. We also examined KLF4 methylated target sites that exhibit medium methylation levels (e.g., ~50%) based on the published methylome dataset. Importantly, KLF4 ChIP-bisulfite sequencing analyses showed that the methylation level of CCCpGCC consensus was significantly increased from 30% and 55% of the genomics inputs to 65% (p=0.001) and 73% (p=0.04) in the ChIP'ed samples for two examined regions, respectively (L4 and L5 in *Figure 4F*; upper panel). As predicted, those nearby CpG sites not embedded in the KLF4 consensus did not show any significant increase in methylation (*Figure 4F*; lower panel). Together, the above analyses provided direct evidence that KLF4 preferentially binds to CCmCpGCC consensus sequence in vivo.

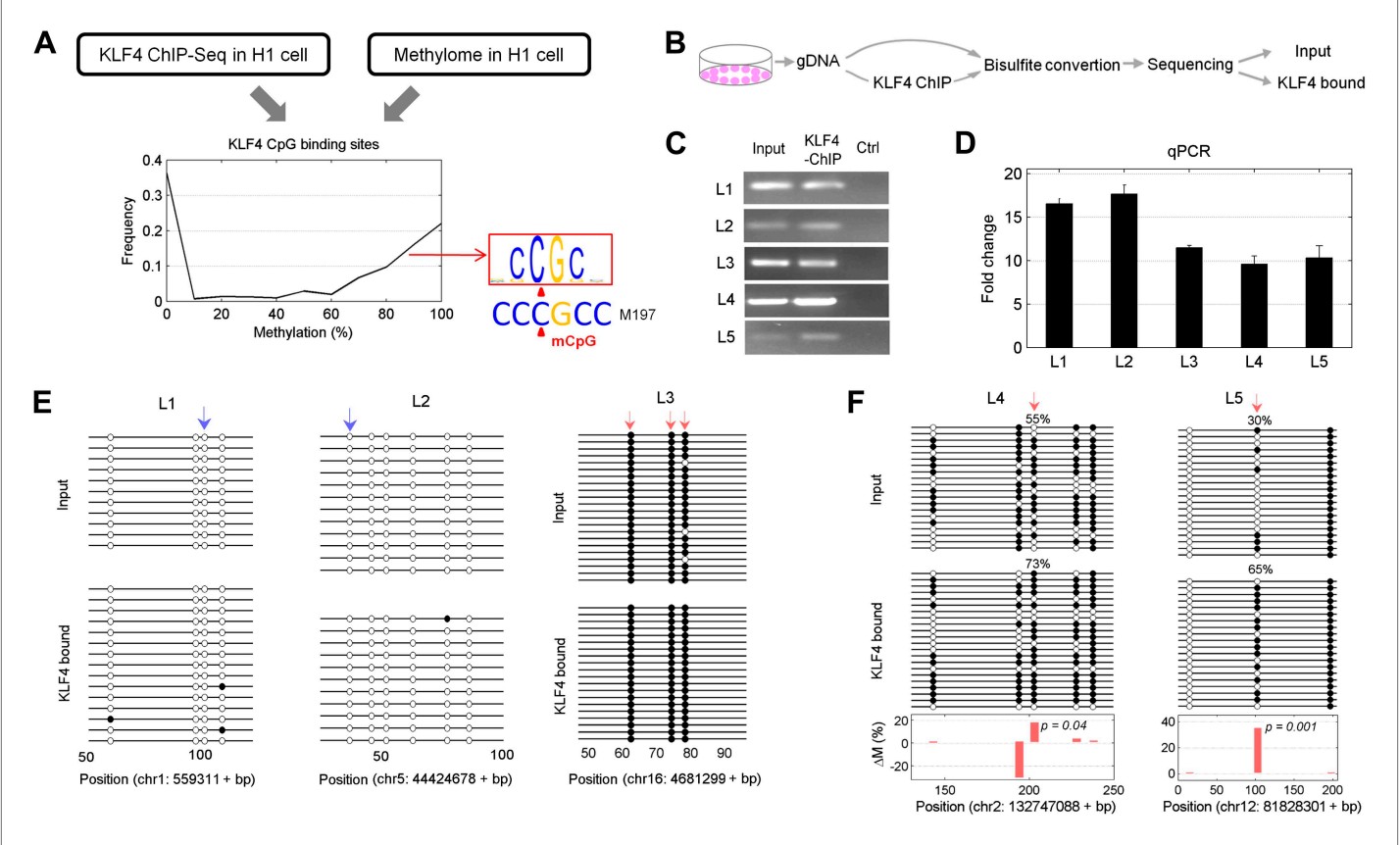

**Figure 4**. Endogenous KLF4 binds to methylated loci in human embryonic stem cells (H1) in vivo. (**A**) Bioinformatics analysis to derive methylated DNA motif logo binding to KLF4 by integrating of KLF4 ChIP-Seq and methylome data in H1 cells. Based on the distribution of methylation level at the KLF4 binding sites, a top methylated consensus motif boxed in red was discovered in the highly methylated sites. As a comparison, M197 sequence recognized by KLF4 in the protein microarray assays is shown below. (**B**) Experimental procedure of KLF4 ChIP-bisulfite sequencing to confirm that KLF4 preferentially interacts with hyper-methylated motifs in H1 cells. (**C**) The gel images of KLF4 ChIP'ed loci (L1: chr1: 559311-559516; L2: chr5: 44424678-44424792; L3: chr16: 4681299-4681481; L4: chr2: 132747088-132747377; L5: chr12: 81828301-81828506) demonstrate specific and direct binding of KLF4 to its target regions. Negative controls were performed in the absence of the anti-KLF4 monoclonal antibodies. (**D**) Analysis of KLF4-ChIP against the five loci using the quantitative real-time PCR (qPCR) method. Fold change at each locus was obtained by taking the ratio of KLF4-ChIP qPCR signals over the negative control signals. Statistics analysis was based on three technical replicates. (**E**) Sanger bisulfite sequencing reads of input and KLF4-ChIP'ed DNA. Filled and blank circles indicate methylated and unmethylated CpG sites, respectively. Blue and red arrows indicate CpGs in the context of motifs M412 and M917, respectively. (**F**) For relatively lower methylation input, KLF4 methylated binding sites tend to have a higher methylation level after KLF4 ChIP. The lower panel in (**F**) shows the methylation differences at each CpG site between the input and KLF4 ChIP'ed DNA. p values were determined by binominal probability density function.

The following figure supplements are available for figure 4:

**Figure supplement 1**. Integration of KLF4 ChIP-seq and methylome data in H1 cell.

**Figure supplement 2**. Five selected KLF4-binding loci for further analyses.

**Figure supplement 3**. An example of KLF4 ChIP-bisulfite sequencing assay.

## Discussion

In this study we have identified numerous human TFs across various subfamilies that showed mCpG- and sequence-dependent binding activity, a much more prevalent phenomenon than previously appreciated (*Karlsson et al., 2008*; *Rishi et al., 2010*; *Quenneville et al., 2011*; *Liu et al., 2012*). It is likely that more proteins with the similar activity are yet to be discovered (*Figure 1—figure supplement 8*), because (1) we only surveyed a tiny fraction of methylated CpG space; (2) we did not screen for TFs that can recognize methylated CHG or CHH sequences in this study, which are present in several cell types

including ESCs, induced pluirpotent stem cells, germline and neural cells (*Ramsahoye et al., 2000*; *Ziller et al., 2011*); (3) we did not include in the screen any hemimethylated DNA motifs, which occurs in vivo on newly synthesized DNAs during replication; and (4) some of the unconventional DNA-binding proteins we discovered in an earlier study might also encode this property (*Hu et al., 2009*).

We estimated false positive and false negative rate for our protein microarray assays. Since 8 of the 11 methylated DNA-binding TFs were confirmed by EMSAs, we estimated that this protein microarray strategy might produce ~27% of false positives. Our approach also showed some false negatives because some known methylated DNA binding proteins were not identified in this study (*Mann et al., 2013*; *Spruijt et al., 2013*). There are several possible reasons to explain this discrepancy. First, since we employed a competition assay on the human TF protein microarrays, only those that preferentially bind to methylated DNA motifs will be identified. In other words, if a protein binds to a DNA sequence irrespective of its methylation status, it will not show strong signals in our screen. Second, the number of DNA sequences we tested is rather small. Based on our simulation analysis (*Figure 1—figure supplement 8*), we expect that there exist more such human TFs yet to be discovered with more DNA motifs tested. Third, because we do not have any TF heterodimers on our arrays, it may also generate false negatives. For example, a recent study (*Spruijt et al., 2013*) employed a generic methylated DNA probe to pull down potential methylated DNA-binding proteins in mouse cell lysates, resulted in the identification of 19 proteins. Among them, 11 have human orthologs and are presented on our human protein microarrays. Five of them, namely MeCP2, NRF1, MBD1/4, and KLF4, showed methylated-specific binding activities in our study. The other six proteins showed weak binding signals in our competition assay, presumably due to the reasons discussed above (*Figure 1—figure supplement 1E*).

Regardless of all the above limitations, we still discovered many novel methylation-dependent DNA-binding activities. For example, Znf114 and Znf416 were identified as new generic methylated-DNA binding proteins, with binding activities to almost all methylated DNA motifs tested in this study. The reason that they were not discovered in previous studies might be due to the fact that most of the previous studies were not performed in a systematic way. In our systematic survey, we identified a total of 47 novel methylated DNA-binding proteins, which significantly expanded the methylation-dependent protein–DNA interaction landscape.

Our findings provide a new framework to better understand mechanisms of DNA methylation-dependent regulation of gene expression, especially in cancer, stem cell and development biology. For example, recent DNA methylome studies in cancer have identified dynamic epigenetic changes, resulting in global reprogramming in gene expression (*Jones and Baylin, 2007*; *Figueroa et al., 2010*; *Noushmehr et al., 2010*). However, CpG methylation does not always correlate with transcription repression in cancers (*Everhard et al., 2009*). Our unbiased screen of the entire human TF family identified many sequence-specific mCpG 'readers', which can presumably interpret the DNA methylation changes and in turn, regulate gene expression in a dynamic environment. Therefore, by presenting distinct binding sites for TFs, methylated cytosines may serve as the fifth alphabet that changes the landscape of TF–DNA interactions and increases the complexity and diversity of gene regulation.

## Materials and methods

### Protein annotation

The protein annotation of transcription factors and cofactors was obtained from our previous study (*Hu et al., 2009*) with some minor manual corrections. The information about TF binding domains was obtained from the Pfam (*Bateman et al., 2004*). The oncogenes and tumor repressor genes were downloaded from UniProtKB (http://www.uniprot.org/). A gene was categorized as a development gene if it has a GO (gene ontology) function which includes a keyword of 'development'. *Supplementary file 1B* lists all transcription factors and co-factors on our protein microarray.

### CpG Methylation of DNA probes and luciferase reporter constructs

Double-stranded DNA probes were generated according to a protocol described previously (*Hu et al., 2011*). Methylated CpG probes or luciferase reporter constructs were prepared as previously described (*Guo et al., 2011*). Briefly, 1 µg of DNA were incubated with 1 µl of *M.Sss*I CpG methyltransferase (4 U/µl, NEB, Ipswich, MA), 1 µl of S-adenosylmethionine (SAM, 32 mM) and 2 µl of 10×NEBuffer 2 in a 20 µl reaction volume at 37°C overnight, followed by incubation with freshly added 1 µl of M.*Sss*I (4 U/µl), 1 µl of SAM (32 mM), 0.5 µl of 10xNEBuffer 2 and 2.5 µl of water at 37°C for 4 hr, then at

65°C for 20 min. The methylation status was confirmed by bisulfite sequencing (*Figure 1—figure supplement 7*).

## Protein microarrays

Human proteins were purified from yeast as GST fusion and arrayed on FAST slides (Whatman, Maidstone, UK) in duplicate as described previously (*Hu et al., 2009*). The protein microarrays were probed with Cy5-labeled methylated DNA motifs in the presence of 10-fold unlabeled (cold), unmethylated competitors using a similar protocol described previously (*Hu et al., 2009*). We selected 154 DNA motifs (*Supplementary file 1A*) and performed tests in duplicate protein microarrays. The slides were then washed and scanned with a GenePix 4000B scanner (Molecular Devices, Sunnyvale, CA) and the binding signals were acquired using the GenePix 6.0 software.

## Protein microarray analysis

We used GenePix 6.0 to align the spot-calling grid and record the foreground and background intensities for every protein spot. *Figure 1—figure supplement 1* shows detailed workflow for protein microarray analysis. The raw binding intensity ($R_{ij}$) of each probe was defined as $F_{ij}/B_{ij}$, where $F_{ij}$ and $B_{ij}$ are the median values of foreground and background signals of the probe at site ($i,j$) on the microarray, respectively. We first normalized the raw signal of each probe based on median value of raw signals of its neighboring probes determined by the window size (window size = 9 × 9 in our study, *Figure 1—figure supplement 1B*). Most probes showed no binding signal ($R' = 1$) and had variations around $R' = 1$ due to background noise distribution as seen in *Figure 1—figure supplement 1C*. In order to evaluate the background noise of each microarray, we selected the shadow part ($N_1$: $R'$, in *Figure 1—figure supplement 1D*) and artificially combined with its symmetrical part around 1 ($N_2$: = $2 - N_1$) to obtain the standard deviation (SD) of the noise distribution ($N = N_1 + N_2$, *Figure 1—figure supplement 1D*). Then the Z-score of each probe was calculated by

$$Z_{i,j} = \frac{R'_{i,j} - \overline{N}}{std(N)},$$

where $R'_{i,j}$ is the locally normalized intensity of probe ($i, j$) on the microarray, $\overline{N}$ and $std(N)$ are mean value and standard deviation, respectively, of noise distribution on the microarray. Since each protein is printed in duplicate on a microarray and each motif binding assay was performed in duplicate, a protein was identified as a positive hit only when all of its four spots produced a Z-score ≥ 3. *Supplementary file 1C* lists all transcription factor and cofactor hits showing binding to at least one methylated DNA motif.

## EMSA

Each binding reaction was carried out with 100 fmol of biotinylated dsDNA probe and 1 pmol of purified protein in 20 µl of binding buffer as described previously (*Hu et al., 2009*). 10-fold (1 pmol) of unlabeled (cold) DNA motifs were added in the competition assays. All the expression clones for proteins used in EMSA were verified by DNA sequencing.

## $K_D$ measurement of transcription factors binding 5mCpG motifs

Binding affinity between a transcriptional factor and a methylated DNA motif was determined by the oblique incidence reflection difference (OIRD) method (*Landry et al., 2004*; *Zhu et al., 2007*; *Fei et al., 2011*). Synthesized DNA motifs were methylated using *Sss*I enzyme as described above ('Materials and methods'), then printed together with their unmethylated counterparts onto Superamine 2 slides (ArrayIt, Sunnyvale, CA) at a concentration of 2.5 µM in 50% DMSO, exposed to 600 µJ UV for 3 min, and dried in a 37°C incubator for 1 hr. We selected ZMYM3, TFAP2A and KLF4 to measure their affinity. We also used MBD2b (Millipore, Billerica, MA) as a bench marker. For each OIRD measurement, a DNA motif slide was first washed with EMSA binding buffer (*Hu et al., 2009*) at 3 ml/min for 5 min, and then flooded with the corresponding TF protein solution at various concentrations as indicated in *Figure 2—figure supplement 3*. The observed on curve was then measured in real time until the OIRD signals reached saturation. To determine the off curve, the EMSA buffer was then pumped through the reaction chamber at a speed of 200 µl/min until the OIRD signals stabilized. Binding and dissociation signals were recorded every 10 s. Data analysis was carried out using Origin 9.0 following Majka J and Speck C's method (*Majka and Speck, 2007*).

## Luciferase assay

The CpG-free luciferase reporter vector used in our study, pCpGL, was a gift from Dr Michael Rehli (*Klug and Rehli, 2006*). Eight units of a DNA motif were subcloned into pCpGL promoter region, and the resulting plasmids (pCpGL-8X-motif) were grown in *Escherichia coli* PIR 1 strain using Zeocin as a selection marker. The CpG sites in the motifs were methylated using M.*Sss*I as described previously. GT1-7 cells were co-transfected with three constructs: pCpGL-8X-motif, pCAGIG expressing the corresponding TF constructs at various concentrations, and pTK-RL (Promega, Madison, WI) using the FugeneHD reagent (Roche, Basel, Switzerland). Cells were harvested 48 hr post-transfection for luciferase reporter assay using the Dual-Luciferase reporter assay system (Promega). All assays were performed in triplicate.

## Methylated DNA binding consensus

We reason that the methylated CpG is the key and conserved position for binding because the unmethylated counterpart shows no binding. Therefore, we first aligned the motif sequences around methylated CpG. For each TF binding to multiple methylated motifs, we then hierarchically clustered all 6-mer binding sequences centered on mCpG. With a selected cutoff for sequence similarity, all the binding motifs were separated into several groups. Each group of sequences was combined to derive the consensus sequences. The consensus sequence for the largest sequence group is shown in *Figure 2A*. If a TF binds to only one motif, the consensus sequence is the same as the binding motif.

## Known DNA binding consensus

We collected the known consensus sequences for the TFs from the literature and databases (TRANSFAC and JASPAR). Some TFs have multiple known consensus sequences. In such cases, we prefer consensus sequences derived from in vitro interaction assays to those derived from in vivo experiments such as ChIP-seq, because the binding sequences derived from ChIP-seq could include methylated sequences. For example, one of the KLF4 consensus sequences in TRANSFAC was based on a ChIP-seq experiment (*Chen et al., 2008*), while another KLF4 binding motif was based on an in vitro oligonucleotide library (*Shields and Yang, 1998*). The known consensus for KLF4 shown in *Figure 2A* was created based on the dataset from in vitro oligonucleotide library using MDscan (*Liu et al., 2002*).

## Site-directed mutagenesis

Site-directed mutagenesis was carried out using the QuikChange Multi Site-Directed Mutagenesis Kit (Agilent, Santa Clara, CA).

## KLF4 methylated binding motifs in vivo

We integrated human methylome data and KLF4 ChIP-seq data to obtain the KLF4 binding landscape in H1 human embryonic stem cells (*Lister et al., 2009*). First, we adopted one popular software MACS (Model-based Analysis for ChIP-Seq) to determine KLF4 binding peaks and summits. The lengths of peaks determined by MACS were from 152 bp to 6062 bp. We selected 95% of these peaks which were shorter than 371 bp for further analysis. To remove the effect of sequence composition difference in genomic regions, we focused on the binding peaks but extended the peaks to ±50 bp to achieve enough 'background' sequences. The 'foreground' was sequences within ±60 bp around KLF4 binding summits. Since we are interested in motifs that could be affected by methylation, we obtained the occurrences of all possible 6-mers having CpG at the center position, e.g. AACGCT. The 6-mers significantly enriched in highly methylated summits (M ≥ 0.8) than in the background were predicted as KLF4 methylated binding motifs. The p-value was calculated based on hypergeometric cumulative distribution function. At the next step, we clustered significant 6-mer sequences (p<0.01 after Bonferroni multiple-test correction, *Supplementary file 1D*) from methylated group to obtain KLF4 methylated DNA binding consensus motif. One consensus sequence is shown in *Figure 4A*.

## H1 cell culture

For feeder-free culture of H1, cells were cultured on Matrigel (BD, 1:80 dilution)-coated plates with MEF-condition media (D-MEM/F12 supplemented with 20% KSR, 2 mM L-glutamine, 100 mM MEM NEAA, and 100 mM beta-mercaptoethanol) supplemented with 4 ng/ml bFGF as previously described (*Chiang et al., 2011*). Media were changed daily.

## Chromatin immunoprecipitation

Chromatin immunoprecipitation (ChIP) was carried out in H1 cells using a rabbit anti-KLF4 antibody (H180; Santa Cruz, Dallas, TX) according to a protocol described previously (*Nelson et al., 2006*), except that the protein A-Sepharose was replaced with Dynabeads Protein A (Life Technologies, Grand Island, NY). Normal rabbit IgG was used for mock IP as a negative control. Primers shown in *Supplementary file 1E* were used in ChIP-PCR and qPCR to detect the binding and enrichment of KlF4 regulatory regions after ChIP.

## Assessment of CpG methylation status by bisulfite sequencing

Sanger bisulfite sequencing was performed as previously described (*Guo et al., 2011*; *Ma et al., 2009*). Purified genomic DNA or ChIP DNA from H1 cells were treated by EZ DNA Methylation-Direct Kit (Zymo Research, Irvine, CA). After bisulfite conversion, regions of interest were PCR-amplified. The primers used in *Figure 4D,E* were listed in *Supplementary file 1E*. PCR products were gel-purified and cloned into a TA vector (Life Technologies). Individual clones were sequenced and aligned with the reference sequence.

## Acknowledgements

We thank Drs Samie Jaffrey, Gary Stormo, Phil Cole, Junjie Guo, Kim Christian, and Zhi Xie for their comments and suggestions. We also thank Dr Michael Rehli for providing the luciferase reporter vector.

## Additional information

### Funding

| Funder | Grant reference number | Author |
| --- | --- | --- |
| National Institutes of Health | R01 GM076102, U54 RR020839, U24 CA160036, U54 HG006434 | Heng Zhu |
| National Institutes of Health | R21 EY021897 | Jiang Qian |
| National Institutes of Health | NS047344, MH087874, ES021957 | Hongjun Song |
| The Simons Foundation Autism Research Initiative | | Hongjun Song |
| National Institutes of Health | NS048271, HD069184 | Guo-Li Ming |
| Dr Miriam and Sheldon G Adelson Medical Research Foundation | | Guo-Li Ming |
| Maryland Stem Cell Research Fund | 2009-MSCRFE-0063-00, 2012-MSCRFII-0063-00 | Guo-Li Ming |

The funders had no role in study design, data collection and interpretation, or the decision to submit the work for publication.

### Author contributions

SH, YS, HNN, JS, Acquisition of data, Analysis and interpretation of data, Drafting or revising the article; JW, G-LM, Analysis and interpretation of data, Drafting or revising the article; QS, YZ, Acquisition of data, Analysis and interpretation of data; EC, HSR, CW, Acquisition of data, Drafting or revising the article; SX, Conception and design, Acquisition of data; SL, HL, Conception and design, Acquisition of data, Analysis and interpretation of data; HW, HS, JQ, HZ, Conception and design, Analysis and interpretation of data, Drafting or revising the article

## Additional files

### Supplementary files

• Supplementary file 1. (**A**) 154 CpG-containing motifs tested on our protein microarray. (**B**) List of transcription factors and cofactors available on our protein microarray. (**C**) Transcription factors

and cofactors binding to methylated DNA motif(s). (**D**) KLF4 binding methylated 6-mers with CpG at the center position obtained by integrating KLF4 ChIP-Seq and methylome data in human H1 cell. (**E**) Information of loci (L1–L5) tested in *Figure 4C–F*: genome locations (hg18), sequences, ChIP PCR and bisulfite-sequencing primers.

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
