## [Decision Letter]

Thank you for sending your work entitled “DNA methylation presents distinct binding sites for human transcription factors” for consideration at *eLife*. Your article has been evaluated by a Senior editor, Detlef Weigel, and 3 reviewers, one of whom is a member of our Board of Reviewing Editors.

The Reviewing editor and the other referees discussed their critiques and request that you respond in writing to the major concerns below.

This manuscript presents an observation with important implications for the field of chromatin regulation and transcription but it was not successful in convincing us of the physiological relevance of the findings; specifically, the ChIP experiments must be improved. The authors should better demonstrate that the dual preference of KLF4 in vivo before this work could be published. The specific points to address are:

1) Figure 4: because of the poor quality (or presentation) of the ChIP-seq data in Figure 4, ChIP-qPCR data for this experiment should be presented here to convince the reader that KLF4 is actually bound to these sites in vivo.

2) The authors should discuss the protein microarray approach in more detail. Do they have any estimations about false positive or false negative hits in their screening?

3) The authors generate a number of KLF4 point mutants that are defective for meCpG binding (R458A, D460A) but still interact with un-methylated DNA (Figure 3). It would be important to dissect which domain or which KLF4 zinc finger is specifically mediating the interaction with un-methylated DNA.

4) Figure 2: this figure lacks negative controls. What happens when adding both methylated and un-methylated competitor DNA? This should abolish KLF4 and TFAP2A binding.

5) For a couple of transcription factor-meCpG interactions, it would be important to determine the *K*_D_ of the interaction using conventional biochemistry and compare these *K*_D_ s to other published meCpG-protein affinities.

6) The authors should discuss their results in the context of previously published literature and discuss any overlap, discrepancies, and novelties. For example: why were ZNF114 and ZNF416 not picked up in previous interactions screenings? Why did the authors not identify meCpG dependent interactions for Rfx proteins?

---

## [Author Response]

*1)*
Figure 4*: because of the poor quality (or presentation) of the ChIP-seq data in 4A, ChIP-qPCR data for this experiment should be presented here to convince the reader that KLF4 is actually bound to these sites* in vivo.

Figure 4 is a schematic plot to demonstrate how we identified the methylated binding logos for KLF4 using the published KLF4 ChIP-seq data in H1 cell (Lister et al., *Nature* 2009). For the sake of simplicity, Figure 4 has been moved to Figure 4—figure supplement 1. In the revised Figure 4, we provide KLF4 ChIP-qPCR data in the form of gel images and quantitative analysis. In the revised Figure 4, we provide evidence that KLF4 lands on un-methylated loci in the sequence context of Motif M412.

*2) The authors should discuss the protein microarray approach in more detail. Do they have any estimations about false positive or false negative hits in their screening*?

We have added more details both to the main text and the Materials and methods section about the protein microarray analysis. We also estimated the false positive rate of the protein microarray assays based on the EMSA results. However, it is hard for us to evaluate the false negative rate by comparing with other results published so far. The reasons are as below: 1) Some proteins identified by others may not present on our protein microarrays, or they came from different species, e.g., mice in the study by Spruijt et al. (“Dynamic Readers for 5-(Hydroxy)Methylcytosine and Its Oxidized Derivatives”, *Cell*. 2013 Feb 28;152(5):1146-59). 2) Our competition assays on the protein microarrays were designed to identify those TFs that prefer methylated DNA motifs, which is very different from other works already published. 3) We only surveyed a limited space of methylated DNA sequences. (So far, we had only tested 154 methylated DNA motifs.) 4) The cutoff of Z-scores used to identify specific methylated DNA-binding proteins was set pretty high. In Figure 1—figure supplement 1, we illustrated the Z-scores for some mouse proteins identified by Spruijt et al., which are just below our cutoff. We have added discussion of false positive and false negative hits in the Discussion section.

*3) The authors generate a number of KLF4 point mutants that are defective for meCpG binding (R458A, D460A) but still interact with un-methylated DNA (*Figure 3*). It would be important to dissect which domain or which KLF4 zinc finger is specifically mediating the interaction with un-methylated DNA*.

We have generated a truncation mutant and identified that it is the typical zf-C2H2 domain in KLF4 that mediates the interaction with the un-methylated DNA motif M412. We have described this new result in the main text and added it in as Figure 3—figure supplement 3.

*4)*
Figure 2*: this figure lacks negative controls. What happens when adding both methylated and un-methylated competitor DNA? This should abolish KLF4 and TFAP2A binding*.

We have performed the negative control experiments by adding both methylated and un-methylated DNA motifs as competitor DNA as recommended. The results are exactly the same as the reviewers predicted (Figure 2—figure supplement 2).

*5) For a couple of transcription factor-meCpG interactions, it would be important to determine the* K_*D*_
*of the interaction using conventional biochemistry and compare these* K_*D*_
*s to other published meCpG-protein affinities*.

We employed an oblique incidence reflectivity difference (OIRD) system (23; 46; 11) to determine binding affinity (i.e., *K*_D_ values) of ZMYM3, TFAP2A, and KLF4 with their corresponding methylated DNA motifs M203, 213, and M197, respectively. As a comparison, a well known methylated DNA-binding protein, MBD2b, was also included. Because the OIRD system, similar to the SPR method, can monitor binding events in a real-time, label-free fashion, we therefore obtained the *K*_on_ and *K*_off_ values, and determined the *K*_D_ values of ZMYM3, TFAP2A, and KLF4 as 460 nM, 399 nM, and 479 nM, respectively, using their newly identified methylated DNA motifs (Figure 2—figure supplement 3). As expected, none of them showed any significant binding activity to the un-methylated DNA motifs in the OIRD measurements, confirming our previous observations. On the other hand, the *K*_D_ values of MBD2b measured against the same motifs are in close range to the three TFs tested, suggesting that these TFs could bind to methylated DNA motifs almost as equally well as MBD2b (Figure 2—figure supplement 3). Finally, the *K*_D_ values of MBD2b measured with the OIRD system is almost the same as reported (i.e., 330 nM; Yinni Yu et al., Direct DNA methylation profiling using methyl binding domain proteins. *Anal. Chem*. 2010, 82, 5012–5019). These new results and a description of the OIRD experiments can now be found in the main text, Figure 2—figure supplement 4, and the Materials and methods section.

*6) The authors should discuss their results in the context of previously published literature and discuss any overlap, discrepancies and novelties. For example: why were ZNF114 and ZNF416 not picked up in previous interactions screenings? Why did the authors not identify meCpG dependent interactions for Rfx proteins*?

ZNF114 and ZNF416 were identified as generic methylated DNA-binding proteins in our study. The reason that they were not discovered in previous studies might be due to the fact that most of the previous studies were not performed in a systematic way. In our systematic survey, we identified a total of 47 novel methylated DNA-binding proteins, which significantly expanded the methylation-dependent protein-DNA interaction landscape. Secondly, a recent study by Spruijt et al. (Spruijt et al. *Cell* 2013) used a mass spectrometry approach to systematically identify methylated DNA-binding proteins. However, their approach was applied to investigate the mouse proteome, which does not encode any ZNF114 and ZNF416 orthologs. On the other hand, we did not identify Rfx proteins as methylated DNA-binding proteins. As we discussed in point 2, there are several reasons for missing previous known methylated DNA-binding proteins in our assay. For example, in our study we used a competition assay, which would not identify those TFs that binds to both methylated and un-methylated DNA motifs equally well. Only the proteins that are preferentially binding to methylated DNA motifs would be identified as hits in our study. Please note that four of the six Rfx proteins on the TF protein microarrays have a Z-score above 2, indicating that they have weak preference to the methylated DNA motifs tested in this study (Figure 1—figure supplement 1). We have added discussion on these issues to the Discussion section.